# Asthma-Associated Long TSLP Inhibits the Production of IgA

**DOI:** 10.3390/ijms22073592

**Published:** 2021-03-30

**Authors:** Dorianne van Heerden, Robert S. van Binnendijk, Samantha A. M. Tromp, Huub F. J. Savelkoul, R. J. Joost van Neerven, Gerco den Hartog

**Affiliations:** 1Cell Biology and Immunology Group, Wageningen University, 6700 AH Wageningen, The Netherlands; doriannevanheerden@gmail.com (D.v.H.); huub.savelkoul@wur.nl (H.F.J.S.); joost.vanneerven@wur.nl (R.J.J.v.N.); 2Center for Immunology of Infectious Diseases and Vaccination, National Institute for Public Health and the Environment, 3720 BA Bilthoven, The Netherlands; rob.van.binnendijk@rivm.nl (R.S.v.B.); s.a.tromp@amsterdamumc.nl (S.A.M.T.); 3Infection and Immunity Department, Amsterdam UMC, 1105 AZ Amsterdam, The Netherlands

**Keywords:** short and long TSLP, asthma, IgA, memory B cells, IL-4, IL-13, IgE, IgG4

## Abstract

Thymic stromal lymphopoietin (TSLP) contributes to asthmatic disease. The concentrations of protective IgA may be reduced in the respiratory tract of asthma patients. We investigated how homeostatic short TSLP (shTSLP) and asthma-associated long TSLP (loTSLP) regulate IgA production. B cells from healthy donors were stimulated in the presence or absence of shTSLP or loTSLP; the concentrations of IgA, IgM, IgE, and IgG antibodies were determined in cell culture supernatants; and B cells were analyzed by flow cytometry. LoTSLP, but not shTSLP, suppressed the secretion of IgA but not of IgE. The type 2 cytokine IL-4, which in addition to loTSLP contributes to asthmatic disease, did not affect the production of IgA or the frequency of IgA+ B cells. Instead, IL-4 increased IgG production, especially of the subclasses IgG2 and IgG4. LoTSLP inhibited IgA secretion by sorted memory B cells but not by naïve B cells. Although loTSLP inhibited IgA production, the vitamin A metabolite retinoic acid promoted the secretion of IgA, also in the presence of loTSLP, suggesting that vitamin A may promote IgA production in asthma. Our data demonstrate that asthma-associated loTSLP negatively regulates the secretion of IgA, which may negatively impact the surveillance of mucosal surfaces in asthma.

## 1. Introduction

Asthma can develop at a young age, with prevalence estimates varying from 1.7–13.5% at the age of 4 years in Europe [1]. Asthma results in significant loss of quality of life and can even lead to mortality. As a chronic disease, asthma needs to be managed well to prevent progression. The optimal management of asthma is dependent on a broad understanding of the biological mechanisms involved in asthma. The airway epithelium is important in initiating initial host defense, and several abnormalities in the epithelial barrier were described in asthma [2]. Important mechanisms identified are the role of Type 2 cytokines such as IL-5, IL-13, and IL-4, and the association with a predisposition to produce IgE type antibodies. More recently, epithelial cytokines such as IL-33 and Thymic stromal lymphopoietin (TSLP) have been implicated the drive the development and/or exacerbation of asthma [3,4].

Respiratory infections with respiratory syncytial viruses (RSV) and rhinoviruses are an important risk factor for the development and exacerbation of asthma, as well as chronic obstructive pulmonary disease (COPD) [5,6,7]. Antibodies of the IgA type, which constitute the predominant antibody at mucosal surfaces, mostly act in a non-inflammatory fashion and are believed to have the ability the neutralize viruses. Upon infection with respiratory viruses, IgA antibodies are induced and RSV-, or other virus-specific IgA levels, are inversely associated with the risk of infection [8,9,10]. In addition to viral infections, the exposure to allergens and subsequent sensitization is a risk factor for asthma disease. Various reports indicate that allergen-specific IgA levels may be reduced in (allergic) asthma patients [11,12]. Relatively lower levels of IgA may either be a consequence of established asthma or contribute to an increased risk of developing asthma due to diminished neutralization of allergens or viruses. IgA antibodies are predominantly produced in mucosal tissues such as the respiratory tract, and IgA is secreted into the lumen of the airways where they are believed to contribute to protection against a broad range of infections and asthma exacerbations as a result of allergen exposure [13,14].

The epithelial-derived cytokine TSLP is involved in causing T helper 2-mediated airway inflammation [15]. The fact that TSLP is an important cytokine that contributes to asthma has been demonstrated in clinical trials where blocking TSLP resulted in alleviation of asthma symptoms [16,17,18]. TSLP is produced in a short (shTSLP) form of 63 amino acids under homeostatic conditions and in a long form (loTSLP) of 159 amino acids following viral infection and in asthma [19]. Interestingly, whereas shTSLP exerts anti-inflammatory activity such as suppressing IL-12 production by DCs and promoting the development of T regulatory cells, loTSLP promotes inflammatory responses such as IFN-γ production [19]. The two forms are not the result of alternative splicing but are regulated by two distinct promotors. TSLP produced by respiratory epithelial cells has been proposed to condition dendritic cells to promote IgA production by B cells [20,21,22]. Although T cells commonly induce antibody production by B cells, the induction of IgA production by TSLP and dendritic cells appeared to be T-cell-independent. Additionally, the cytokines A proliferation-inducing ligand (APRIL) and B-cell activating factor (BAFF) that are produced in the local mucosa could together with IL-10 induce the production of IgA, especially IgA2 [23,24,25]. Additionally, the vitamin A metabolite retinoic acid (RA) is a potent inducer of (IgA) antibody-secreting cells (CD38+CD20-) and was found to reduce allergic inflammation [24,26,27].

The latter data were derived in TSLP receptor knockout mice and have not yet been confirmed in humans. Importantly, the differential regulation of IgA by human B cells by the short versus the long form of TSLP has not yet been investigated. Therefore, we aimed to elucidate whether homeostatic shTSLP, or inflammatory loTSLP as expressed in the respiratory epithelium at elevated levels in asthma, differentially affect the production of IgA by B cells.

## 2. Results

### 2.1. The Effect of TSLP on IgA Production

B cells from healthy donors were stimulated under T cell-dependent conditions using irradiated CD40L-expressing L cells and cytokines. The addition of shTSLP or loTSLP did not affect B cell proliferation, either in the presence or absence of CpG (Figure 1A). However, loTSLP but not shTSLP significantly reduced the secretion of IgA (Figure 1B). ShTSLP or loTSLP did not seem to affect the production of IgE, which was highly variable between donors (Figure 1C).

Infection of primary normal human bronchial epithelial cells with three different RSV isolates induced the secretion of APRIL and BAFF (Figure 1D). Similar to T cell-dependent inhibition of IgA production, loTSLP consistently, albeit not significantly, showed an inhibition of the production of IgA under T cell-independent conditions using APRIL or BAFF and IL-10 (Figure 1E).

### 2.2. Regulation of IgA and IgG Production by the Type 2 Cytokines IL-4 and IL-13

B cells were stimulated in a T cell-dependent manner in the presence of the type 2 cytokines IL-13 and IL-4. IL-13 and IL-4 alone or in combination of either of the forms of TSLP did not alter the levels of IgM or IgA (Figure 2A,B). IL-13 and IL-4 promoted the production of IgG, regardless of the presence of either form of TSLP (Figure 2C–F). Especially, the induction of IgG2 and IgG4 following stimulation with IL-4 was notable. IgG4, normally the least produced IgG subclass, was produced at levels similar to IgG1 when stimulated with IL-4. shTSLP and loTSLP did not alter the effect of IL-4 on concentrations of IgG1-4 observed. Although type 2 cytokines are known to induce IgE production, no significantly elevated levels were observed in any of the conditions (high donor variability, data not shown).

Flow cytometric analysis of the frequencies of cell surface expression of IgA and IgG by B cells revealed that stimulation with IL-4 resulted in increased numbers of IgG-positive B cells (Figure 3). Although loTSLP or shTSLP alone did not significantly increase the percentage of IgG B cells, together with IL-13 and especially IL-4 the proportion of IgG B cells strongly increased, which confirmed our observations in the cell culture supernatants. Of the two TSLP forms, especially co-stimulation with shTSLP and IL-4 or IL-13 increased the frequency of IgG+ B cells. The proportion of IgA B cells was not significantly altered by IL-13, IL-4, or either form of TSLP. Since membrane IgA expression did not change, no correlation between IgA in supernatants and IgA on the membrane of B cells was observed. However, membrane expression of IgG correlated significantly with concentrations of IgG1 in the cell culture supernatants (*p* = 0.003).

### 2.3. FACS Sorting Shows That TSLP Regulates IgA Production by Memory B Cells

To investigate whether TSLP would affect naïve (CD3-CD19+CD27-IgD+) and memory (CD3-CD19+CD27+IgD-) B cells differently, both populations were separated by fluorescence-activated sorting (>99% pure) from PBMCs and then stimulated in a T cell-dependent manner in the absence or presence of either form of TSLP. As expected, naïve B cells showed the lowest IgA and IgG production levels, while memory B cells secreted around 10-fold higher levels of IgG and IgA (Figure 4). LoTSLP but not shTSLP tended to suppress production of IgA by memory B cells. IgG production by memory B cells was not altered by TSLP.

### 2.4. Restoration of loTSLP-Suppressed IgA Production by the Vitamin A Metabolite Retinoic Acid

TSLP, either short or long, did not influence the differentiation of B cells into antibody secreting cells (Figure 5A,B). In line with our previous results, RA upregulated IgA production but not IgG1 production, and it did so in the presence of loTSLP (Figure 5C,D).

## 3. Discussion

Here, we show that loTSLP but not shTSLP inhibits the production of IgA by memory B cells. The effect of loTSLP was selective for IgA, and was not observed for IgM, IgE, or IgG1-4. Retinoic acid also promotes the production of IgA in the presence of loTSLP and may thus be able to restore IgA production in asthma patients in the presence of aberrant TSLP signaling.

A previous study showing the involvement of TSLP in regulating the production of IgA used a TSLP-receptor knockout mouse model, and B cells were not directly stimulated with TSLP but indirectly via DCs [22]. B cell responses vary considerably between mice and humans, and these studies did not discriminate between short and long TSLP. Another study in patients with immunoglobulin A nephropathy found a positive association between tonsillar TSLP expression and IgA production [28]. In these patients with immunoglobulin A nephropathy APRIL, BAFF and TGF-ß were also increased and could be causally related to the elevated levels of IgA [23,24,28]. These studies indicate that the role of TSLP in regulating IgA production may be more complicated than currently understood and warrants further research. Additional research is also needed to better understand the regulation of production of shTSLP and loTSLP.

Based on our findings, we propose that altered expression of loTSLP may drive diminished IgA production in asthma patients. The effect of TSLP may also be different for the production of IgA1 and IgA2, which is worth further study since especially IgA2 levels may be diminished in patients suffering from asthma and eczema [12]. An additional mechanism that may contribute to diminished IgA levels on the luminal surface of the respiratory epithelium is the inhibition of pIgR-mediated transport of IgA by IL-4 and IL-13 [29]. Since we previously observed diminished allergen-specific IgA levels in eczema and asthma patients [12], and that aberrant allergen-specific and virus-specific IgA responses may even precede asthma development, future studies could investigate whether restoration of IgA production may help to protect asthma patients against viral infections and associated exacerbations [12,30]. Here, we used B cells from healthy donors with unknown atopy status, and therefore our results may be influenced by the presence of atopy in some of the donors used, which may also explain the extensive variation in the IgE levels in the cell culture supernatants. Additional insight into the role of TSLP in regulating IgA production could be obtained using B cells from asthma patients. Our data showing that retinoic acid promotes IgA production in the presence of loTSLP is also promising, and the beneficial effect of retinoic acid is more profound than the inhibitory effect of loTSLP. These data suggest that asthma patients may benefit from vitamin A supplementation. This is further strengthened by other studies showing that retinoic acid limits airway inflammation in asthma patients as well as suppresses inflammation in other diseases [24,26,31]. Vitamin A deficiency is relatively rare in western countries, but a few studies in communities with higher prevalence of vitamin A deficiency from Nepal and India found an association between low vitamin A concentrations in serum and asthma in children [32,33].

The induction of the inflammatory form of TSLP is also observed in COPD, eczema, eosinophilic esophagitis, and inflammatory bowel disease [34,35,36,37,38]. In addition, TSLP has been implicated in exacerbating airway inflammation induced by exposure to aeroallergens [39], and diesel exhaust particles [40] and TSLP may be (partly) responsible for exacerbations of asthma upon exposure to environmental pollutants [41]. These studies did not discriminate between loTSLP and shTSLP, and there are no studies investigating whether loTSLP results in altered IgA production in these diseases or exposures. However, in COPD patients’ elevated levels of IgA were observed to be associated with increased production of APRIL, BAFF, and IL-6 in these patients and could be reproduced by coculturing B cells with respiratory epithelial cells of COPD patients [25]. To this end, it may be relevant to also evaluate the influence of single nucleotide polymorphisms of TSLP found to be associated with asthma and other diseases where TSLP is involved [37,42,43,44,45]. TSLP can be observed in circulation, which makes it suitable as biomarker, and the fact that TSLP can be observed in circulation rather than only locally in the mucosa may indicate that inhibition of IgA production by loTSLP can reach beyond the local environment [46,47]. Since therapeutics targeting TSLP are available, studies in human patients receiving TSLP antagonists could investigate whether this results in recovery of IgA production.

In conclusion, we show that the long and short form of TSLP differentially regulate IgA production, which may help explain the mechanisms behind the development and exacerbations of asthma and support the effectiveness of therapeutic interventions targeting aberrant TSLP production.

## 4. Materials and Methods

### 4.1. B Cell Isolation and Stimulation

Blood samples were used from generally healthy donors, and atopy was not used as an exclusion criterium. Peripheral blood mononuclear cells (PBMC) were isolated using Ficoll density centrifugation, and cryopreserved PBMC from healthy donors were thawed and B cells were sorted using the CD19 EasySep kit (#17854, StemCell technologies, Vancouver, BC, Canada) according to the manufacturers’ descriptions. 1000 B cells per well were added to round-bottom 96 wells plates and stimulated for 5 days, followed by a medium change and a second stimulation mixture (Table 1). Two approaches were used to culture B cells: (a) mimicking T cell-dependent (TD) stimulation by stimulating with irradiated CD40L-transfected fibroblasts and (b) mimicking T cell-independent stimulation by co-stimulating with the mucosal cytokines APRIL or BAFF. Media were changed at day 5 with slightly altered stimulation conditions. The first 5 days cells were stimulated with the factors indicated under ‘’added day 0’′ in Table 1. At day 5, half of the media were replaced and cells subsequently stimulated with the factors indicated under “added day 5” [24,48]. The donors used throughout the experiments and figures are indicated in Appendix A.

Blood samples were obtained from the Dutch blood bank Sanquin and a vaccination cohort approved under METC no NL37852.094.11. All voluntary donors provided informed consent following the guidelines set by the Dutch government. Personal data processed in this study did not require evaluation by a medical ethical committee.

### 4.2. Analysis of APRIL, BAFF and Ig Concentrations

Primary bronchial epithelial cells of three different donors were cultured and infected with RSV as described [49]. Cell-free culture supernatants were collected at day 7 post infection and analyzed using the Legendplex B cell activator panel (#740535, Biolegend, San Diego, CA, USA). From the B cell cultures, cell-free supernatants were collected at day 11 when maximum production of antibodies was observed based on previous experiments and stored at −20 °C until analyses. IgG1, IgG2, IgG3, IgG4, IgM, and IgA concentrations were determined using the Legendplex 6-plex (#740640, BioLegend) and IgE using the Legendplex single-plex assay (#740641).

For all Legendplex assays 5 µL of beads, detecting antibodies, assay buffer, and prediluted cell culture supernatants were added to V-shaped polypropylene 96-well plates and incubated for 2 h, while shaking (750 rpm) at room temperature in the dark. After 2 h, 5 µL streptavidin-PE was added and mixed and incubated for an additional 30 min as before. Then, 200 µL wash buffer was added, and following centrifugation (5 min 500× *g*) supernatants were removed and 100 µL wash buffer was added. Three hundred events per bead region were acquired on a Fortessa X-20 (BD Biosciences, San Jose, CA, USA). Data were fitted using a five-parameter logistic fit, and samples outside the limits of quantitation were retested at adjusted dilutions.

### 4.3. Flow Cytometry

Cells were harvested at day 7 and washed in FACS buffer (0.5% *w*/*v* BSA and 2 mM EDTA in PBS). Cells were incubated with antibody mix for 30 min at 4 °C. Following washing, cells were resuspended in FACS buffer and acquired on a BD Fortessa X-20 and analyzed using FlowJo. The following antibodies were used for flow cytometric analyses: CD19, CD20, CD27, CD38, IgA, IgG, and IgD, as detailed in Table 2.

### 4.4. Cell Sorting

Cryopreserved PBMCs were stained for CD3, CD19, IgD, and CD27. Using a BD Melody, CD3-CD19+ cells were sorted into naive (CD3-CD19+CD27-IgD+) and memory B cells (CD3-CD19+CD27+IgD-) and cultured as described above.

### 4.5. Statistical Analyses

Data were entered into Graphpad prism. Concentrations were log-transformed prior to statistical testing using ANOVA with Sidaks correction for multiple testing. Figures show mean values and standard errors of the mean. Test results *p* < 0.10 are shown unless indicated otherwise.

## Figures and Tables

**Figure 1 ijms-22-03592-f001:**
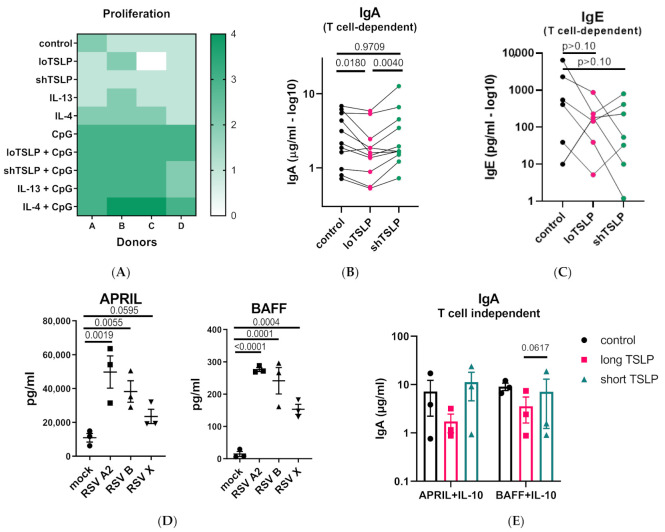
Regulation of production of IgA by B cells stimulated under T cell-dependent or T cell-independent conditions. (**A**) The viability of B cells of 4 donors stimulated under T cell-dependent conditions using irradiated CD40L-expressing L cells in the absence or presence of CpG, and cytokines (thymic stromal lymphopoietin (TSLP), IL-4, and IL-13) for 5 days. (**B**) The secretion of IgA in supernatants of B cells cultures at day 11 after stimulation in the presence or absence of long TSLP (loTSLP) or short TSLP (shTSLP) (11 donors, 4 independent experiments). Statistical test results of paired ANOVA analysis on log-transformed data are shown. (**C**) For six of the donors of two independent experiments stimulated in ‘’B’’, IgE levels were also determined (all *p*-values > 0.10). (**D**) Production of APRIL and BAFF by air-liquid differentiated normal human bronchial epithelial cells from three donors following infection with three different strains of respiratory syncytial viruses (RSV) for 1 week. Statistical differences are calculated on log-transformed concentrations using One-Way ANOVA. (**E**) IgA production in supernatants of B cells cultured for 11 days under T cell-independent conditions in the absence or presence of TSLP for three donors in a single experiment.

**Figure 2 ijms-22-03592-f002:**
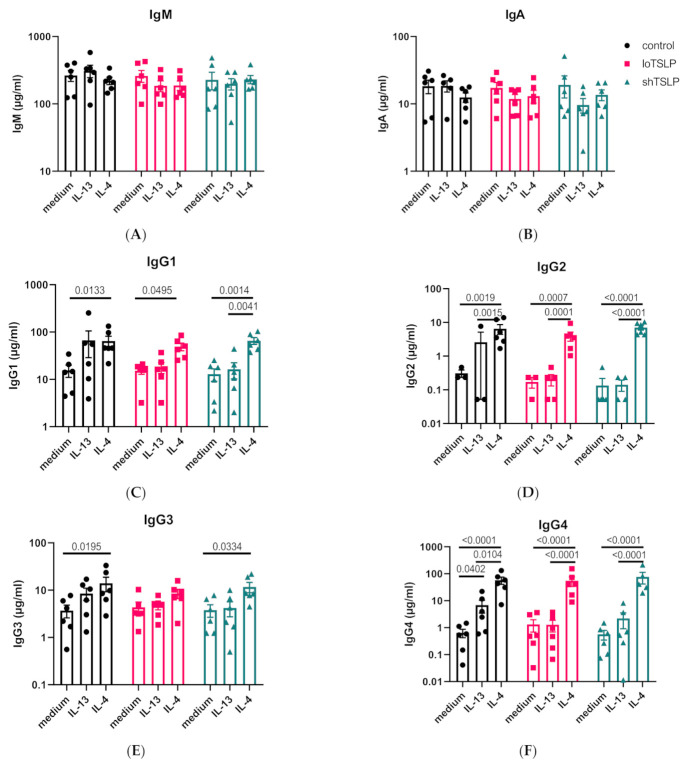
Regulation of Ig secretion by IL-4, IL-13, and TSLP. The concentrations of IgM (**A**), IgA (**B**), IgG1 (**C**), IgG2 (**D**), IgG3 (**E**), and IgG4 (**F**) in B cell culture supernatants after 11 days of stimulation using the T cell-dependent protocol using irradiated CD40L expressing cells. Mean concentrations and standard error of the mean are shown. Statistical test results are from ANOVA with Sidaks correction for multiple testing using log-transformed concentration data of four to six donors and two independent experiments.

**Figure 3 ijms-22-03592-f003:**
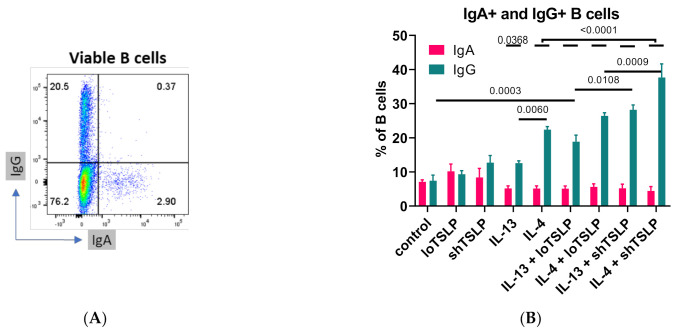
Flow cytometric analysis of B cells after 7 days of stimulation. An example of IgG and IgA flow cytometric analysis (**A**) and summarized data showing mean and standard error of the mean (**B**) of 4–5 donors are shown from two independent experiments. The overlaying bar indicates that the differences of the five small bars are all *p* < 0.0001. ANOVA with Sidaks correction for multiple testing was used.

**Figure 4 ijms-22-03592-f004:**
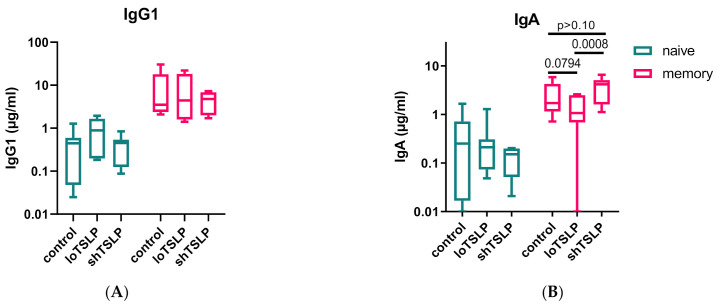
IgG and IgA production by naïve and memory B cells stimulated with loTSLP or shTSLP. Naïve and memory B cells were stimulated for 11 days using the T cell dependent protocol and supernatants analyzed for concentrations of IgG1 (**A**) and IgA (**B**). Boxplots show mean, second, and third (box) and first and fourth percentile of seven donors tested in two independent experiments.

**Figure 5 ijms-22-03592-f005:**
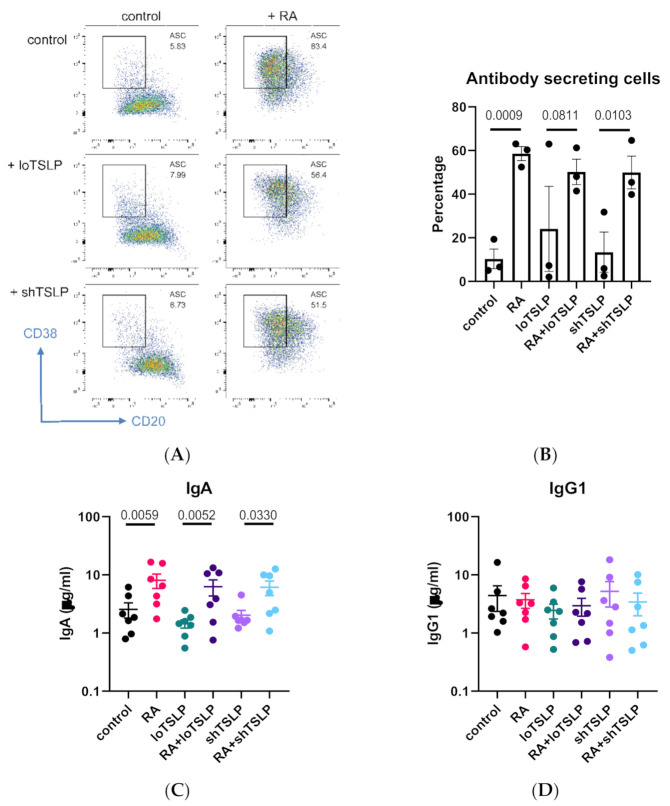
The role of retinoic acid (RA) in inducing secretion of IgA. CD19 B cells were cultured for 7 days and stained for CD20 and CD38 (**A**) in the presence or absence of RA and TSLP (**B**) for 3 donors in a single experiment. B cells were cultured for 11 days and supernatants analyzed for concentrations of IgA (**C**) and IgG1 (**D**) for seven donors in two independent experiments. Graphs show mean and standard error of the mean, and statistical test results are obtained from ANOVA with Sidaks correction for multiple testing.

**Table 1 ijms-22-03592-t001:** Stimuli added at day 0 and 5 and used concentrations.

Stimulus			Added	
	**Concentration**	**Supplier**	**Day 0**	**Day 5**
**T Cell Dependent Cell Culture**				
**CD40L**	500 irradiated CD40L cells per well	RIVM	X	
**CpG (ODN 2006)**	3 µg/mL	Invivogen	X	
**IL-2**	10 ng/mL	R&D Systems	X	X
**IL-21**	100 ng/mL	Myltenyi Biotec		X
**IL-10**	10 ng/mL	BD Biosciences	X	X
**T Cell Independent Cell Culture**				
**APRIL**	200 ng/mL	Peprotech	X	X
**BAFF**	200 ng/mL	Peprotech	X	X
**IL-10**	10 ng/mL	BD Biosciences	X	X
**Stimuli Investigated for Effect on IgA Production**				
**Short TSLP**	10 ng/mL	Maria Rescignio/Giulia Fornasa, HUNIMED, Italy *	X	X
**Long TSLP**	10 ng/mL	BioLegend	X	X
**IL-13**	10 ng/mL	Peprotech	X	X
**IL-4**	10 ng/mL	Peprotech	X	X
**RA**	100 µM	Sigma Aldrich		X

* reference [18].

**Table 2 ijms-22-03592-t002:** Antibodies used for flow cytometric analyses and fluorescence activated cell sorting.

Target Protein	Fluorochrome	Clone	Manufacturer
**FACS Cell Analyses**			
CD19	BV786	SJ25C1	BD Biosciences, San Jose, CA, USA
CD20	BV510	2H7	BioLegend, San Diego, CA, USA
CD27	BV711	M-T271	BioLegend
CD38	APC-H7	HB7	BD Biosciences
IgA	FITC or APC	IS11-8E10	Myltenyi Biotec, Bergisch-Gladbach, Germany
IgG	BV421	M1310G05	BioLegend
**FACS Cell Sorting**			
CD3	PerCP-Cy5.5	UCHT1	BioLegend
CD19	PE-Cy7	HIB19	BioLegend
IgD	FITC or AF700	IA6-2	BioLegend
CD27	BV711	M-T271	BioLegend

## Data Availability

The data presented in this study are available in this article.

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
