# Peer review of "Asthma-Associated Long TSLP Inhibits the Production of IgA"

_ijms, 2021, doi:10.3390/ijms22073592_

Round 1
Reviewer 1 Report
The paper by van Heerden and colleagues is an exploratory study on the effects of epithelial-derived TSLP on IgA production, suggesting that long TSLP can inhibit IgA production by cultured memory B cells. In contrast, they confirm that retinoic acid promotes IgA production. Thus study is of interest in the field of immunoregulation given the importance of both TSLP and IgA in mucosal homeostasis and disease. There are however important issues that should be addressed to reinforce the data and their interpretation.
Major issues.
- Methodology: the authors should mention whether non-atopic donors were selected as blood cell donors (or unselected; a panel of IgE to pneumallergens could be done on their stored serum if available) as this may have impacted the results. The authors mentionned using freshly isolated PBMCs (abstract) vs cryopreserved PBMCs (methods), which is inconsistent. The authors should also clari the rationale for the cytokine cocktail used for the 2 step process in T-dependent stimulation of B cells (4.1). Finally, why selecting 6 donors for the IgE assays (and not assess the 11 donors as for IgA)?The aurors integrate to their B-cell data one experiment using ALI-cultured epithelial cells and their release of BAFF and APRIL (Fig 1D). The reviewer wonders whether it would be posible (and more consistent) to assay TSLP in those supernatants.
- The authors integrated an experiment addessing BAFF and APRIL release by ALcultured epithelial cells. the reviewer wonders whether TSLP should also be assayed for consistency.
- The major finding, namely the inhibition of IgA prooduction by long TSLP (Fig 1) is not reproduced in Figure In addition, surface IgA data did not show IgA downregulation by TSLP (only trend for memory cells). Those data in contrast to those (vvery clear) with retinoic acid, should prompt the authors to ascertain their observations in order to drive more convincingly their main message. Titration of both IgA1 and IgA2 could maybe help in that purpose
- Minor issues.
- Better explain the biological difference betweence long and short TSLP (ref 18).
- Focus on research data in the Results section.
- References 13 and 23are incomplete. The authors could also discuss the work of Ladjemi et al. on B cell production of IgA in asthma (AJRCCM 2018) and COPD (Eur Respir J 2014).
- Retinoic acid is mentioned twice in Table 1.
Reviewer 2 Report
The study is interesting, well-designed, and novel; the manuscript is well-written.
The investigated issue is significant since tezepelumab, an anti-TSLP monoclonal antibody, has been demonstrated to be effective in treating eosinophilic and non-eosinophilic severe asthma and likely will be soon available for clinical practice. Thus, presented research pointed at the additional mode of its action.
I have only some minor comments:
Introduction:
1) The abbreviation of TSLP needs to be explained when it appears for the first time in the text.
2) Please explain the difference between long and short TSLP - requires an explanation for those not involved in the TSLP research, e.g., clinicists.
3) TSLP is an epithelial cell alarmin, also produced after environmental harm particles exposure. Do we know what kind of TSLP is secreted in that situation, lo- or shTSLP? Please extend the manuscript to include that issue. It might be important in the context of more severe respiratory tract infections reported in urbanized areas.
Results:
4)Line 77: ”B cells from healthy donors were stimulated under T cell-dependent conditions using CD40L-expressing L cells”. – Please explain what “L-cells” means.
5)Line 101: “…using irradiated CD40L expressing cells..” – please also clarify here what kind of cells were used.
6)The impact of TSLP on IgE production varied a lot among donors. Do you have any idea why it has happened? Did you check IgE level or eosinophilia in donors' peripheral blood?
Materials:
7)Please clarify how many donors were used for each series of experiments and in how many copies experiments were performed. The numbers of donors have been provided in Fig. 1 description, albeit they should also be given in the Materials section. Moreover, it is unclear whether the different B-cell experiments were performed on the same donors. If the donors were the same, please also provide whether it was any relationship among the results of various experiments in a particular donor, e.g, FACS and cell culture.
8)Line 228 “Primary bronchial epithelial cells were cultured..”– from how many donors? The number (n=3) was provided in Fig description, albeit it should also be given here. Were supernatants pooled? – please clarified.
